# Changes in Probiotic Lachnospiraceae Genera Across Different Stages of COVID-19: A Meta-Analysis of 16S rRNA Microbial Data

**DOI:** 10.3390/microorganisms13092061

**Published:** 2025-09-04

**Authors:** Clarissa Reginato Taufer, Juliana da Silva, Pabulo Henrique Rampelotto

**Affiliations:** 1Graduate Program in Genetics and Molecular Biology, Universidade Federal do Rio Grande do Sul, Porto Alegre 91501-970, Braziljuliana.silva@unilasalle.edu.br (J.d.S.); 2Graduate Program in Health and Human Development, Universidade La Salle, Canoas 92010-000, Brazil; 3Bioinformatics and Biostatistics Core Facility, Instituto de Ciências Básicas da Saúde, Universidade Federal do Rio Grande do Sul, Porto Alegre 91501-970, Brazil

**Keywords:** Lachnospiraceae, microbiome, gut microbiota, COVID-19, disease severity

## Abstract

The gut microbiome has emerged as a potential modulator of COVID-19 severity, and there is particular interest in the Lachnospiraceae family due to its role in maintaining gut homeostasis. This study presents a comprehensive meta-analysis of microbiome datasets from multiple investigations focused on gut microbiota across various stages of COVID-19. We used a standardized bioinformatics pipeline based on Mothur v.1.47.0 and the SILVA v.138 reference database to analyze 16S rRNA gene sequencing data targeting the V3-V4 regions. Our findings reveal consistent patterns of depletion in key Lachnospiraceae genera, particularly *Lachnospira* and *Roseburia*, correlating with increased COVID-19 severity. Complex patterns were observed for *Blautia* and *Coprococcus*, suggesting strain-specific responses to disease states. In addition, several unclassified Lachnospiraceae taxa showed differential abundance across severity levels, indicating the need for further characterization of these potentially important bacteria. These results provide robust evidence for the association between specific Lachnospiraceae genera and COVID-19 severity. The observed microbial shifts suggest potential mechanisms by which gut dysbiosis may contribute to COVID-19 pathogenesis, including reduced production of beneficial metabolites and compromised intestinal barrier function. These findings highlight the potential of Lachnospiraceae genera as biomarkers for COVID-19 severity and suggest avenues for developing targeted probiotic interventions in COVID-19 management.

## 1. Introduction

The human gut microbiome, a complex ecosystem of microorganisms residing in the gastrointestinal tract, plays a major role in maintaining human health. Among the diverse microbial communities inhabiting the gut, the Lachnospiraceae family (class Clostridia, phylum Firmicutes) is a significant contributor to gut homeostasis and overall well-being. This family comprises more than 100 genera, including well-characterized members such as *Blautia*, *Coprococcus*, *Lachnospira*, and *Roseburia*, along with numerous taxa that lack standing in nomenclature [1].

The Lachnospiraceae family represents one of the most abundant bacterial groups in the healthy human gut microbiota, constituting approximately 10–45% of the total bacterial population in feces [2]. These obligate anaerobic bacteria exhibit remarkable metabolic versatility [3], particularly in their ability to ferment a wide range of carbohydrates, including complex polysaccharides like starch and inulin [4]. This metabolic capacity not only contributes to the efficient extraction of energy from dietary components but also results in the production of essential metabolites, most notably short-chain fatty acids (SCFAs) such as butyrate and acetate [5].

Beyond their metabolic functions, Lachnospiraceae members contribute significantly to host immunity and defense mechanisms. These bacteria facilitate colonization resistance against pathogenic microorganisms through various mechanisms, including the production of antimicrobial compounds such as antibiotics and the modulation of local pH through acid production [3,6,7]. The immunomodulatory effects of Lachnospiraceae extend to their involvement in intestinal inflammatory processes, where they help maintain a balanced immune response [8].

Alterations in Lachnospiraceae populations have been associated with various pathological conditions, including reduced abundance of these bacteria in patients with type 2 diabetes, inflammatory bowel disease (IBD), obesity, and specific types of cancers [9,10,11,12]. These associations suggest that maintaining healthy populations of Lachnospiraceae may be essential for preventing or managing these conditions.

Recently, the role of the gut microbiome in infectious diseases has been increasingly recognized, particularly in the context of the COVID-19 pandemic. Several studies have reported alterations in the gut microbiota composition of COVID-19 patients, with a notable reduction in beneficial bacteria, including members of the Lachnospiraceae family [13,14,15,16]. These findings have led to investigations into the potential causality between gut microbiome composition and COVID-19 susceptibility [17].

The relationship between Lachnospiraceae abundance and COVID-19 severity has emerged as a particularly intriguing area of study, with correlations between reduced Lachnospiraceae populations and increased disease severity [13]. Some genera within this family, such as *Lachnospira* and *Blautia*, have increased in post-COVID-19 patients [18], suggesting a potential role in recovery. However, other members of the family remain reduced in post-COVID-19 patients, indicating that alterations in the gut microbiome may persist even after the acute phase of the disease [19].

The potential mechanisms by which Lachnospiraceae may influence COVID-19 outcomes are multifaceted. The production of SCFAs by these bacteria may play a role in modulating the immune response and maintaining the integrity of the intestinal barrier, which could be essential in preventing the translocation of the SARS-CoV-2 virus or its components into the bloodstream [20,21]. In addition, the immunomodulatory effects of Lachnospiraceae may help regulate the inflammatory response, potentially reducing the cytokine storm associated with severe COVID-19 cases [22].

Despite the increasing evidence linking Lachnospiraceae to COVID-19, direct comparisons between studies present significant challenges. These difficulties arise from variations in data analysis techniques, reference databases used for microbial identification and categorization, and differences in patient populations and disease severity classifications. To address these issues and gain a more comprehensive understanding of the role of Lachnospiraceae in COVID-19, a standardized meta-analysis of existing datasets is necessary.

This study aims to conduct a systematic re-evaluation of microbiome datasets from investigations focused on gut microbiota across various stages of COVID-19. We hypothesize that a standardized meta-analysis will reveal consistent patterns in Lachnospiraceae abundance across different stages of COVID-19 severity, providing valuable insights into the potential role of these bacteria in disease progression and recovery. Our analysis incorporates 16S rRNA gene sequencing data, specifically targeting the V3-V4 hypervariable regions. This approach was selected for its established reliability and cost-effectiveness in bacterial community profiling [23,24]. The meta-analysis implements a standardized bioinformatics pipeline using Mothur v.1.47.0 for sequence processing and the SILVA v.138 reference database for taxonomic classification, ensuring consistency across all datasets as we have already applied for other key probiotic genera in COVID-19 [25].

These findings may have significant implications for understanding the pathophysiology of COVID-19 and could potentially inform the development of novel therapeutic interventions targeting the gut microbiome [26]. Furthermore, this study contributes to the broader field of microbiome research by demonstrating the value of standardized meta-analysis in reconciling disparate findings across multiple studies. This approach may serve as a model for future meta-analyses in microbiome research, particularly in the context of rapidly evolving fields such as infectious disease research [27].

## 2. Materials and Methods

### 2.1. Study Selection and Obtaining the Sequencing Datasets

Our database search focused on studies of the gut microbiome in COVID-19 patients using 16S rDNA gene sequencing. We prioritized studies with publicly accessible raw datasets in repositories such as the National Center for Biotechnology Information (NCBI) and European Nucleotide Archive (ENA). Our analysis focused exclusively on 16S rRNA sequences from stool samples, specifically targeting the V3-V4 hypervariable region.

To ensure comparability, we included only studies that provided clear definitions of COVID-19 stages (Mild, Moderate, Severe, and Critical). We excluded studies involving therapeutic interventions to minimize potential confounding factors. This rigorous selection process yielded seven studies suitable for our analysis (Table 1).

It is worth noting that one study (PRJNA700830) focused solely on critical cases, which precluded its inclusion in comparative analyses across disease stages. Nevertheless, we incorporated this study’s data into our overall analysis to maximize the breadth of our investigation.

### 2.2. Sequencing Read Data Processing and Analysis

Our sequence analysis followed a previously established protocol [25]. We used Mothur v.1.47.0 for initial processing which included the removal of barcodes and primers from the raw sequences [35]. A stringent quality control pipeline was implemented to ensure data integrity.

Sequences were filtered based on multiple criteria: quality scores (minimum Q30), length constraints (270–300 bp), absence of ambiguous bases, and homopolymer length (≤6 bp). We used VSEARCH to detect and eliminate potential chimeric sequences [36]. To further enhance data quality, we removed singleton sequences, which often represent PCR or sequencing artifacts [37].

Following these preprocessing steps, we classified the refined sequence set using the SILVA v.138 reference database. Subsequently, we clustered the sequences into Amplicon Sequence Variants (ASVs) [38]. To identify significant variations in genus-level abundance across different COVID-19 severity stages, we applied Linear discriminant analysis Effect Size (LEfSe) analysis [39]. Statistical significance was determined using a corrected *p*-value threshold of 0.05.

## 3. Results

In total, 581 samples were analyzed, comprising 142 control samples and 439 COVID-19 samples, including 64 from mild patients, 75 from moderate, 182 from severe, and 96 from critical cases. The overall analysis included both negative and healthy controls (Figure 1). Patients who were hospitalized for other reasons but tested negative for COVID-19 are referred to as negative controls (PRJNA747262).

The WHO’s ‘Clinical Management of COVID-19: Living Guideline’ (World Health Organization, 2023) and the ‘Guidelines for the Diagnosis and Treatment of Coronavirus Disease 2019 (COVID-19) in China’ [40] served as the basis for the severity grading applied in the included studies. With comparable categorization criteria, both classifications include mild, moderate, severe, and critical severities. Mild cases display mild clinical symptoms without any indications of pneumonia, moderate cases show clinical evidence of non-severe pneumonia, severe cases exhibit clinical signs of severe pneumonia, and critical cases, characterized by acute respiratory distress syndrome (ARDS) symptoms, require mechanical ventilation and admission to the intensive care unit.

The studies are from Europe and Asia and cover six countries (Bangladesh, China, Germany, India, Switzerland, and Ireland), while no studies from America or Africa were observed. Two studies used NovaSeq 6000, while four studies used MiSeq as sequencing technology.

The PRJEB50040 study compared healthy controls with COVID-19 cases of varying severity, including fatal cases and severe survivors. Health controls chosen from databases were also used in this investigation. We decided to use just the original controls from the trials to minimize collection and sampling inconsistencies.

The PRJNA767939 study compared samples of COVID-19 patients with original healthy controls without distinguishing between severities. Only this one study was reanalyzed to compare COVID-19 with healthy controls. Both ventilated (ICU) and non-ventilated (non-ICU) patient samples were analyzed in PRJEB61722. To make comparisons, we used the parameters of severity definition to classify ventilated patients as critical and non-ventilated patients as severe.

In PRJNA747262, non-severe (mild + moderate) and severe (severe + critical) cases were compared with positive and negative individuals. Since severity was specified for every sample, microbiome changes across all COVID-19 severities could be assessed. Mild, moderate, and severe cases were assessed in PRJNA684070, whereas healthy controls (original) were compared in PRJNA895415. In our reanalysis, these categories were preserved.

Table 2 presents the differential abundance results using the standard protocol for the genera *Blautia*, *Coprococcus*, *Lachnospira*, and *Roseburia* across studies with different COVID-19 severities, in addition to the findings of the original work. *Blautia* was not mentioned in the text for studies PRJEB50040 and PRJEB61722; however, it was identified as statistically significant in the supplementary material for PRJEB50040 (Mann–Whitney). The study PRJNA895415 identified that this genus was more abundant in mild COVID-19 patients compared to those with severe cases, which is consistent with our findings that also revealed a statistically significant decrease in this genus as disease severity increased. In addition, PRJNA684070 identified *Blautia* as a potential biomarker, noting it as the primary taxon reduced in the feces of COVID-19 patients. However, our analysis did not detect this genus.

The genus *Coprococcus* was identified at lower relative abundance levels in high-risk patients in the study PRJEB50040. Similarly, PRJNA684070 reported a decrease in this genus among COVID-19 patients compared to control groups, suggesting it as a potential biomarker. In our analysis, this genus was not identified as differentially abundant in PRJEB50040 and was not found in PRJNA684070, indicating that the genus *Coprococcus* does not show significant differences when evaluating different disease severities. The studies PRJEB61722 and PRJNA895415 do not report any information on this genus, and our analyses also did not find it to be differentially abundant. The absence of constant statistical significance demonstrates that *Coprococcus* is not a major factor in the development of the illness.

The genus *Lachnospira* was not mentioned in PRJEB50040, PRJEB61722, PRJNA895415, and PRJNA684070. In our analysis, this genus was identified in 4 out of the 5 studies, being differentially abundant in 3 of them, PRJEB50040, PRJEB61722, and PRJNA895415, showing a reduction with increasing severity, potentially indicating a relationship with COVID-19 severity. Notably, this genus was also not identified in PRJNA684070. The consistent reduction in *Lachnospira* with increasing COVID-19 severity suggests a potential link between this genus and disease progression. The lack of identification in the original studies further emphasizes the need for further research to clarify the role of Lachnospira in COVID-19 and its potential as a biomarker for disease severity.

As for the genus Roseburia, it was associated with the low-risk COVID-19 group in PRJEB50040, while PRJEB61722 found this genus to be reduced in ventilated patients compared to non-ventilated ones, and PRJNA747262 identified a lower relative abundance in severe/critical COVID-19 (severe + critical) compared to non-severe COVID-19 (mild + moderate), being one of the two genera capable of differentiating severe from non-severe patients. Meanwhile, PRJNA895415 and PRJNA684979 did not mention this genus. In our reanalysis, *Roseburia* was differentially abundant in PRJEB50040, PRJEB61722, and PRJNA747262, showing a reduction with increasing severity in PRJEB61722 and PRJNA747262 and an increase with increasing severity in PRJEB50040. These contrasting findings highlight the complexity of *Roseburia*’s role in COVID-19, suggesting that its relationship with disease severity may vary depending on specific patient contexts or study parameters.

The reanalysis is presented in Table 3 for PRJNA767939, which compared samples from COVID-19 patients with healthy controls for the genera *Blautia*, *Coprococcus*, *Lachnospira*, and *Roseburia*. The study does not provide information about these genera in the main text. However, all genera were identified in our reanalysis, and the genera *Lachnospira* and *Roseburia* were differentially abundant. Differences in the databases used may be the cause of the difference between the original study and our reanalysis.

As a diverse family with many examples of unclassified genera, in addition to evaluating the main genera of the Lachnospiraceae family, we also assessed other unclassified taxa to understand the relationship between the Lachnospiraceae family and COVID-19. These genera do not present standing names in nomenclature. A total of 14 unclassified genera were identified across the six studies (Table 4 and Table 5). Table 4 presents all the unclassified genera identified in the reanalysis of studies with varying severities of COVID-19. No study mentioned unclassified genera from the Lachnospiraceae family in the original text. However, in PRJEB50040, the genera Lachnospiraceae_NK4A136_group, Lachnospiraceae_UCG.010, and Lachnospiraceae_NC2004_group were cited in the supplementary material as being higher in low-risk patients compared to high-risk patients. This study was also the only one to mention the Lachnospiraceae family as decreased in high-risk patients.

In our analysis, the 11th genera of Lachnospiraceae showed significant results for differential abundance across different severities of COVID-19 (Table 4). These include Lachnospiraceae_FCS020_group for PRJNA895415, Lachnospiraceae_ge for PRJEB50040 and PRJNA895415, Lachnospiraceae_NK4A136_group for PRJEB61722 and PRJNA747262; Lachnospiraceae_NK4B4_group for PRJNA747262; Lachnospiraceae_ND3007_group for PRJNA895415; Lachnospiraceae_UCG_001 for PRJEB61722; Lachnospiraceae_UCG_002 for PRJEB61722; Lachnospiraceae_UCG_004 for PRJEB50040, PRJEB61722, and PRJNA895415; Lachnospiraceae_UCG_006 for PRJEB50040; Lachnospiraceae_UCG_010 for PRJEB50040; and Lachnospiraceae_unclassified for PRJEB61722 and PRJNA684070.

In PRJNA767939, which evaluated controls versus COVID-19, four unclassified genera were identified (Table 5). Of these, only the genus Lachnospiraceae_UCG_004 was differentially abundant in healthy controls, and it was shown to be depleted in COVID-19 patients.

To assess the general alterations of these genera in relation to the severity of COVID-19, we additionally analyzed all samples from the selected studies grouped by severity (Figure 1). These data highlight the distinct patterns that the genera exhibit during the progression of COVID-19 severity. *Coprococcus*, *Roseburia*, and Lachnospiraceae_NK4A136_group demonstrate a decrease with increasing severity, while *Blautia* and Lachnospiraceae_ge show a decrease up to moderate severity and then an increase in severe and critical cases. In contrast, Lachnospiraceae_unclassified shows an increase from mild to moderate and then decreases in severe and critical cases. *Lachnospira* exhibits a distinct pattern, with a decrease in abundance as severity increases but an increase in severe cases.

## 4. Discussion

This comprehensive meta-analysis of microbiome data from multiple studies provides valuable insights into the dynamics of Lachnospiraceae genera across different stages of COVID-19. Our findings reveal consistent patterns in the abundance of specific Lachnospiraceae genera, particularly *Lachnospira* and *Roseburia*, which show significant reductions associated with increased disease severity. These results not only corroborate previous studies but also offer a more refined understanding of the relationship between gut microbiota and COVID-19 progression.

The consistent depletion of *Lachnospira* and *Roseburia* observed across multiple studies suggests a robust association between these genera and disease progression. *Lachnospira*, known for its ability to produce SCFAs, particularly butyrate, plays a major role in maintaining gut barrier integrity and modulating immune responses [5,41]. The reduction in *Lachnospira* in severe COVID-19 cases may lead to decreased SCFA production, potentially compromising the intestinal barrier and allowing for increased translocation of pathogens or their components. This could contribute to the systemic inflammation observed in severe COVID-19 cases [42]. Similarly, *Roseburia*, another important butyrate-producing genus, has been associated with anti-inflammatory properties and the maintenance of gut homeostasis [41]. The observed reduction in *Roseburia* abundance with increasing COVID-19 severity aligns with previous findings in other inflammatory conditions [43]. The loss of *Roseburia* may further exacerbate the inflammatory state in COVID-19 patients, potentially contributing to the cytokine storm observed in severe cases [42].

The relationship between *Blautia* abundance and COVID-19 severity presented a more complex pattern. Our meta-analysis revealed that *Blautia* showed variable responses across different studies, with some indicating reduction in abundance up to moderate severity followed by an increase in severe and critical cases. This variability may reflect the genus’ diverse metabolic capabilities and strain-specific responses to disease states. The genus *Blautia* includes multiple species with distinct metabolic profiles and immunomodulatory properties [8,44]. Some *Blautia* species produce antimicrobial compounds and modulate immune responses through the production of specific metabolites [3]. The observed increase in *Blautia* abundance in severe cases might represent a compensatory mechanism or the proliferation of specific strains better adapted to the altered gut environment during severe COVID-19.

The analysis of *Coprococcus* populations revealed inconsistent patterns across studies, suggesting that its relationship with COVID-19 severity may be influenced by additional factors not captured in our analysis. As *Coprococcus* species play important roles in tryptophan metabolism and the production of anti-inflammatory compounds [45,46], the variable responses observed in our analysis might reflect the complex interactions between host metabolism, immune responses, and microbial communities during COVID-19 progression. These findings highlight the need for more detailed investigations into the strain-level variations and metabolic capabilities of *Coprococcus* species in the context of COVID-19.

Our meta-analysis also identified several unclassified Lachnospiraceae genera that showed differential abundance across COVID-19 severity levels. This finding highlights the vast diversity within the Lachnospiraceae family and indicates the potential relevance of yet-to-be-characterized bacterial taxa in the context of COVID-19. The genus Lachnospiraceae_UCG_004 showed consistent reductions with increasing disease severity across multiple studies, suggesting its potential role as a biomarker for COVID-19 progression. In addition, other unclassified genera such as Lachnospiraceae_NK4A136_group and Lachnospiraceae_ge showed variable patterns across studies but were consistently associated with changes in COVID-19 severity. These findings emphasize the need for more comprehensive genomic and functional characterization of these unclassified taxa to fully understand their potential roles in health and disease [3,10,47].

The presence of several unclassified Lachnospiraceae taxa in our analysis also highlights a significant challenge in microbiome research: the limited resolution of 16S rRNA gene sequencing in identifying bacteria at the species or strain level [3,23]. The limited taxonomic resolution of 16S rRNA sequencing arises from the conserved nature of this gene across many bacterial species, which makes it difficult to distinguish closely related taxa at the species or strain level. While 16S rRNA sequencing is highly effective for identifying bacteria at higher taxonomic levels, such as genus, its reliance on relatively small variable regions within the gene often fails to capture the subtle genetic differences that differentiate species or strains, particularly within diverse and phylogenetically complex taxa like Lachnospiraceae. Most genera from the family Lachnospiraceae contain multiple species that share highly similar 16S rRNA sequences, leading to ambiguous or overlapping classifications when using this method. This limitation is significant because strains within the same species can exhibit vastly different functional roles, such as in metabolism, pathogenicity, or interactions with the host. As a result, relying solely on 16S rRNA sequencing can obscure the ecological and clinical relevance of specific strains, hindering efforts to understand microbial dynamics and their implications for health or disease. This limitation demonstrates the need for more advanced sequencing techniques, such as shotgun metagenomics, to provide a more detailed understanding of the microbial community structure and function in the context of COVID-19 [48,49]. The application of such advanced techniques could reveal additional insights into the specific strains and functional capabilities of Lachnospiraceae that may be particularly relevant to COVID-19 pathogenesis and recovery.

The consistent patterns of Lachnospiraceae depletion observed in our meta-analysis provide valuable insights into the potential mechanisms by which the gut microbiome may influence COVID-19 pathogenesis. The reduction in beneficial bacteria, particularly butyrate-producing genera like *Roseburia* and *Lachnospira*, may contribute to the dysregulation of the immune response and increased intestinal permeability observed in severe COVID-19 cases [50,51]. This dysbiosis could potentially exacerbate the systemic inflammation characteristic of severe COVID-19, creating a vicious cycle of immune dysregulation and microbial imbalance [52].

These findings suggest several potential opportunities for therapeutic interventions. The development of probiotic formulations specifically enriched in Lachnospiraceae genera, particularly *Roseburia* and *Lachnospira*, could help maintain gut homeostasis and modulate immune responses in COVID-19 patients [53]. Targeted probiotic interventions could be designed to restore the balance of beneficial bacteria and potentially mitigate the severity of COVID-19 symptoms. However, it is important to note that the efficacy of such interventions would need to be rigorously evaluated through well-designed clinical trials before implementation.

Dietary interventions aimed at promoting the growth of beneficial Lachnospiraceae members could also be explored as a complementary approach. Supplementation with specific fibers known to promote butyrate production might help maintain the abundance of these bacteria [54]. Such prebiotic interventions could provide a more sustainable approach to modulating the gut microbiome, as they work by selectively promoting the growth of beneficial bacteria already present in the gut. Given the importance of short-chain fatty acids produced by Lachnospiraceae, direct supplementation with butyrate or other beneficial metabolites could also be investigated as a potential therapeutic strategy [55]. This approach could overcome the need for bacterial fermentation and directly provide beneficial metabolites to the gut environment. However, the optimal delivery method and dosage for such metabolite-based therapies would need to be carefully determined to ensure their efficacy and safety.

Our meta-analysis, while providing valuable insights, also has several limitations that should be acknowledged. The use of 16S rRNA gene sequencing, while cost-effective and widely used, has limitations in taxonomic resolution and functional inference. Future studies using shotgun metagenomics could provide more detailed insights into species-level changes and functional capabilities of the gut microbiome in COVID-19 [56,57]. This approach would allow for a more comprehensive understanding of the specific strains and metabolic pathways involved in the observed changes in Lachnospiraceae populations.

The cross-sectional nature of most of the analyzed studies limits our ability to infer causality or track individual-level changes over time. Longitudinal studies following patients from diagnosis through recovery could provide more robust insights into the temporal dynamics of gut microbiome changes in COVID-19 [13,58]. Such studies could help elucidate whether the observed changes in Lachnospiraceae populations are a cause or consequence of disease severity, and how these changes evolve during infection and recovery.

The influence of various confounding factors on the observed patterns cannot be fully excluded from our analysis. Variables such as diet, medication use (particularly antibiotics), and comorbidities may all impact on the composition and function of the gut microbiome. Future studies should aim to control these factors more rigorously. In addition, the studies included in our analysis represent diverse geographical locations and populations. While this diversity strengthens the generalizability of our findings, it may also introduce variability due to differences in dietary habits, environmental factors, and genetic backgrounds.

Our analysis focused primarily on taxonomic changes, but future studies should incorporate metabolomics and metatranscriptomics to provide a more comprehensive understanding of the functional implications of the microbial shifts [59,60]. The integration of multiple omics approaches, including metagenomics, metabolomics, metatranscriptomics, and host genomics, could provide a more complete picture of the interactions between the gut microbiome and host responses in COVID-19 [61,62]. This multi-omics approach would allow for a better understanding of the mechanistic links between Lachnospiraceae populations and disease outcomes.

Mechanistic studies using in vitro and animal models are needed to elucidate the specific pathways by which Lachnospiraceae members influence COVID-19 pathogenesis and immune responses [63]. These studies could help identify specific metabolites or bacterial products that mediate the observed effects and potentially lead to the development of more targeted therapeutic approaches. Clinical trials evaluating the efficacy of targeted probiotic or prebiotic interventions in preventing or mitigating COVID-19 severity are also necessary [64,65].

The long-term impacts of COVID-19 on gut microbiome composition and function, particularly in relation to post-acute sequelae of SARS-CoV-2 infection, represent another important area for future investigation [66,67]. Understanding how changes in Lachnospiraceae populations persist or evolve during long-term recovery could provide insights into the role of the microbiome in post-COVID conditions and potentially inform therapeutic strategies for managing these long-term complications. Large-scale studies in diverse populations could help identify population-specific patterns and potential therapeutic targets [68,69]. Such studies would be particularly valuable in understanding how genetic, dietary, and environmental factors influence the relationship between Lachnospiraceae and COVID-19 outcomes across different populations. This knowledge could lead to more personalized approaches to microbiome-based interventions for COVID-19 management.

## 5. Conclusions

The comprehensive meta-analysis of microbiome data across multiple studies presented in this work provided robust evidence for the association between specific Lachnospiraceae genera and COVID-19 severity. The standardized bioinformatics pipeline used in this study allowed us to compare the findings across studies, leading to broader conclusions about the role of Lachnospiraceae in COVID-19. Our findings reveal consistent patterns of depletion in key genera such as *Lachnospira* and *Roseburia*, which correlate with increased disease severity. These results demonstrate the potential role of the gut microbiome in COVID-19 pathogenesis and highlight promising opportunities for therapeutic interventions. The observed changes in Lachnospiraceae populations offer insights into the complex interaction between gut microbiota, immune responses, and disease outcomes in COVID-19. While our study provides valuable information, it also emphasizes the need for further research, including longitudinal studies, mechanistic investigations, and clinical trials of microbiome-based interventions. Overcoming the challenge of classifying Lachnospiraceae is essential for understanding their functional roles within the gut microbiota, particularly in the context of COVID-19 and other infectious diseases.

## Figures and Tables

**Figure 1 microorganisms-13-02061-f001:**
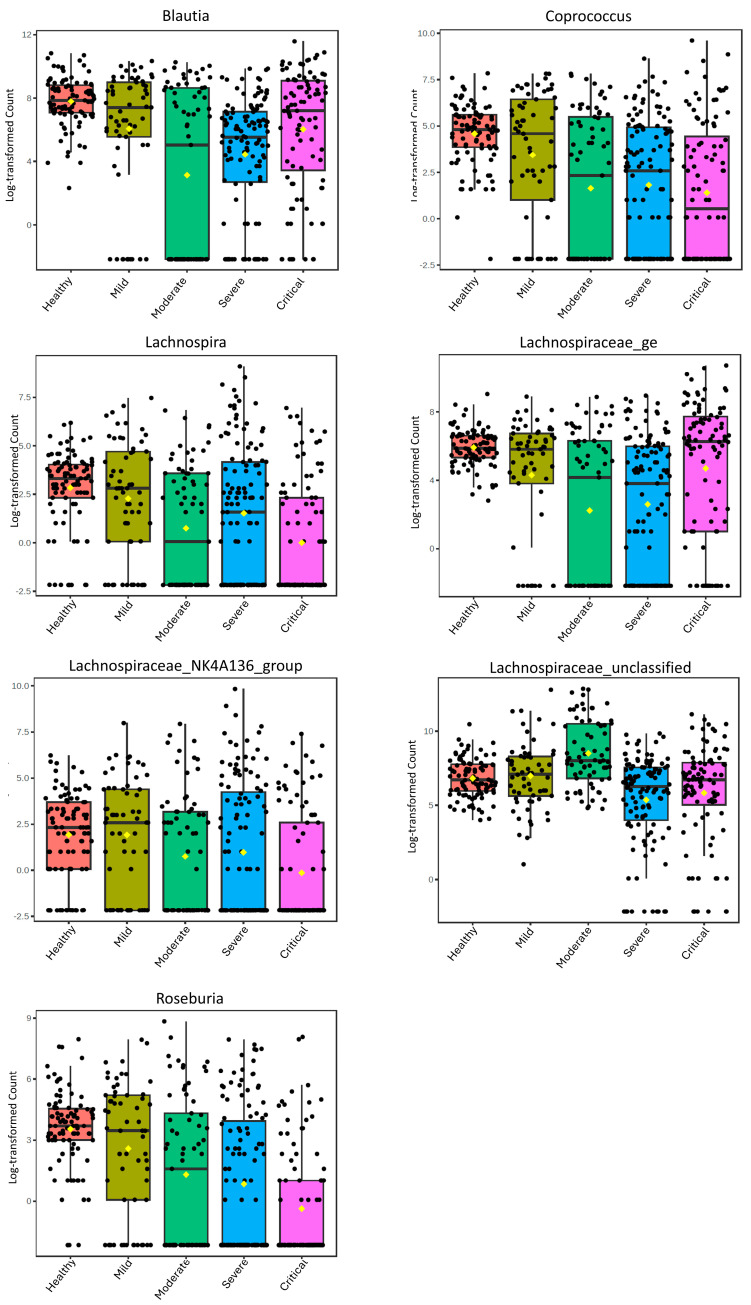
Boxplots of the relative abundance of the Lachnospiraceae Genera in healthy controls and at different levels of severity of COVID-19 with LDA Score > 1.

**Table 1 microorganisms-13-02061-t001:** Paper details identified from the studies included in the reanalysis.

Autor; Year	Accession Number	Country	Type of Study	NGS Technology	N	Groups
Albrich et al., 2022 [28]	PRJEB50040	Switzerland and Ireland	Cohort	MiSeq	98	8 mild, 24 moderate, and 66 severe
Gaibani et al., 2021 [29]	PRJNA700830	Italy	Case–control	MiSeq	69	COVID-19
Galperine et al., 2023 [30]	PRJEB61722	Switzerland	Cohort	MiSeq	57	42 severe, 15 critical
Rafiqul Islam et al., 2022 [31]	PRJNA767939	Bangladesh	Cross-section	MiSeq	37	15 healthy, 22 COVID-19
Reinold et al., 2021 [32]	PRJNA747262	Germany	Cross-section	NovaSeq 6000	212	95 negative, 44 mild, 35 moderate, 26 severe, 12 critical
Talukdar et al., 2023 [33]	PRJNA895415	India	Cohort	MiSeq	52	7 mild, 45 severe
Wu et al., 2021 [34]	PRJNA684070	China	Case–control	NovaSeq 6000	56	32 healthy, 5 mild, 16 moderate, 3 severe

**Table 2 microorganisms-13-02061-t002:** Differential abundance of *Blautia*, *Coprococcus*, *Lachnospira*, and *Roseburia* in each disease stage according to the original study and the re-analysis using a standard protocol. Bold numbers indicate statistical significance. “-” indicates that the genus was not cited; “NR” indicates values not reported.

Genus	Study	Reference Database	Statistical Analysis	*p* Values	FDR	Healthy	Mild	Moderate	Severe	Critical	LDA Score
*Blautia*	PRJEB50040	NR	Mann–Whitney	0.00042096	-	-	-	-	-	-	-
	Standard Protocol	Silva v. 138	Lefse	0.082563	0.49618	-	528.38	568.08	324.02	-	2.09
	PRJEB61722	EzBioCloud	NBZIMM	-	-	-	-	-	-	-	-
	Standard Protocol	Silva v. 138	Lefse	0.67677	0.87127				187.86	272.73	1.64
	PRJNA747262	Greengenes 13.8	Lefse	-	-	-	-	-	-	-	-
	Standard Protocol	Silva v. 138	Lefse	0.27837	0.62404	1093.1	1159.1	1336.7	852.62	1301.8	2.39
	PRJNA895415	Silva v. 138	Lefse	0.0018892	0.027866	-	-	-	-	-	5.21
	Standard Protocol	Silva v. 138	Lefse	**0.036234**	0.37246	-	468.71	-	168.31	-	2.18
	PRJNA684070	Greengenes 13.8	Lefse	NR	NR	NR	NR	NR	NR	NR	NR
	Standard Protocol	Silva v. 138	Lefse	-	-	-	-	-	-	-	-
*Coprococcus*	PRJEB50040	NR	Mann–Whitney	0.003043936	-	-	-	-	-	-	-
	Standard Protocol	Silva v. 138	Lefse	0.11478	0.54002	-	31.5	16.417	15.909	-	0.944
	PRJEB61722	EzBioCloud	NBZIMM	-	-	-	-	-	-	-	-
	Standard Protocol	Silva v. 138	Lefse	0.16682	0.66966	-	-	-	64.762	33	−1.23
	PRJNA747262	Greengenes 13.8	Lefse	-	-	-	-	-	-	-	-
	Standard Protocol	Silva v. 138	Lefse	0.22521	0.59503	122.15	216.5	199.11	152	137.83	1.68
	PRJNA895415	Silva v. 138	Lefse	-	-	-	-	-	-	-	-
	Standard Protocol	Silva v. 138	Lefse	0.18663	0.7538	-	41.429	-	23.978	-	0.988
	PRJNA684070	Greengenes 13.8	Lefse	NR	NR	NR	NR	NR	NR	NR	NR
	Standard Protocol	Silva v. 138	Lefse	-	-	-	-	-	-	-	-
*Lachnospira*	PRJEB50040	NR	Mann–Whitney	7.98309 × 10^−5^	-	-	-	-	-	-	-
	Standard Protocol	Silva v. 138	Lefse	**0.0048226**	0.1136	-	21.875	33.083	11.455	-	1.07
	PRJEB61722	EzBioCloud	NBZIMM	-	-	-	-	-	-	-	-
	Standard Protocol	Silva v. 138	Lefse	**0.0057487**	0.35032	-	-	-	146.17	22	−1.8
	PRJNA747262	Greengenes 13.8	Lefse	-	-	-	-	-	-	-	-
	Standard Protocol	Silva v. 138	Lefse	0.21381	0.58498	42.895	72.091	65.114	54.346	29.083	1.35
	PRJNA895415	Silva v. 138	Lefse	-	-	-	-	-	-	-	-
	Standard Protocol	Silva v. 138	Lefse	**0.012363**	0.21696	-	83.714	-	10.4	-	1.58
	PRJNA684070	Greengenes 13.8	Lefse	-	-	-	-	-	-	-	-
	Standard Protocol	Silva v. 138	Lefse	-	-	-	-	-	-	-	-
*Roseburia*	PRJEB50040	NR	Mann–Whitney	0.000113838	-	-	-	-	-	-	-
	Standard Protocol	Silva v. 138	Lefse	**0.0087579**	0.15472	-	8.625	19.583	20.879	-	0.853
	PRJEB61722	EzBioCloud	NBZIMM	NR	NR	-	-	-	NR	NR	NR
	Standard Protocol	Silva v. 138	Lefse	**0.039976**	0.58486	-	-	-	67.048	4.7333	−1.51
	PRJNA747262	Greengenes 13.8	Lefse	**<0.05**	NR	NR	NR	NR	NR	NR	>3.5
	Standard Protocol	Silva v. 138	Lefse	**0.042058**	0.27312	78.274	119.34	175.77	114.12	43.917	1.83
	PRJNA895415	Silva v. 138	Lefse	-	-	-	-	-	-	-	-
	Standard Protocol	Silva v. 138	Lefse	0.25183	0.76757	-	21.571	-	9.3333	-	0.852
	PRJNA684070	Greengenes 13.8	Lefse	-	-	-	-	-	-	-	-
	Standard Protocol	Silva v. 138	Lefse	-	-	-	-	-	-	-	-

**Table 3 microorganisms-13-02061-t003:** Differential abundance of the genera in Healthy vs. COVID-19 according to the original study and the re-analysis using a standard protocol. Only one study was eligible for this analysis. Bold numbers indicate statistical significance. “-” indicates that the genus was not cited.

Genus	Study	Reference Database	Statistical Analysis	*p* Values	FDR	Healthy	COVID-19	LDA Score
*Blautia*	PRJNA767939	NCBI	Kruskal–Wallis	-	-	-	-	-
	Standard Protocol	Silva v. 138	LEfSe	0.81092	0.89347	18.067	36.045	−1
*Coprococcus*	PRJNA767939	NCBI	Kruskal–Wallis	-	-	-	-	-
	Standard Protocol	Silva v. 138	LEfSe	0.39419	0.57006	3.1333	2.9091	0.0462
*Lachnospira*	PRJNA767939	NCBI	Kruskal–Wallis	-	-	-	-	-
	Standard Protocol	Silva v. 138	LEfSe	**0.035134**	0.12232	9.6	13.091	−0.439
*Roseburia*	PRJNA767939	NCBI	Kruskal–Wallis	-	-	-	-	-
	Standard Protocol	Silva v. 138	LEfSe	**0.001058**	**0.016576**	3.4	0.81818	0.36

**Table 4 microorganisms-13-02061-t004:** Differential abundance of unclassified Lachnospiraceae in each disease stage according to the re-analysis using a standard protocol. Bold numbers indicate statistical significance. “-” indicates that the genus was not cited.

Genus	Study	Reference Database	Statistical Analysis	*p* Values	FDR	Healthy	Mild	Moderate	Severe	Critical	LDA Score
Lachnospiraceae_FCS020_group	PRJEB50040	Silva v. 138	Lefse	0.2352	0.65374	-	4.5	1.2917	1.3485	-	0.416
Lachnospiraceae_FCS020_group	PRJEB61722	Silva v. 138	Lefse	0.27561	0.71539	-	-	-	1.7619	0.8	−0.171
Lachnospiraceae_FCS020_group	PRJNA895415	Silva v. 138	Lefse	**0.09367**	0.61012	-	-	2.2857	0.84444	-	0.236
Lachnospiraceae_ge	PRJEB50040	Silva v. 138	Lefse	**0.0072566**	0.14405	-	80.125	56	50.924	-	1.19
Lachnospiraceae_ge	PRJEB61722	Silva v. 138	Lefse	0.82062	0.911	-	-	-	157.19	126.67	−1.21
Lachnospiraceae_ge	PRJNA747262	Silva v. 138	Lefse	0.094113	0.44042	249.07	307.32	392.63	357.81	640.75	2.29
Lachnospiraceae_ge	PRJNA895415	Silva v. 138	Lefse	**0.012753**	0.21696	-	-	74.857	19.733	-	1.46
Lachnospiraceae_NK3A20_group	PRJNA747262	Silva v. 138	Lefse	0.22463	0.59503	1.5368	1.9773	1.0571	0.038462	0.66667	0.294
Lachnospiraceae_NK4A136_group	PRJEB61722	Silva v. 138	Lefse	**0.0001754**	**0.044201**	-	-	-	155.52	20.267	−1.84
Lachnospiraceae_NK4A136_group	PRJNA747262	Silva v. 138	Lefse	**0.00019874**	**0.010158**	30.895	82	105.94	102.08	68	1.59
Lachnospiraceae_NK4B4_group	PRJNA747262	Silva v. 138	Lefse	**0.00020731**	**0.010158**	0	0.22727	0.17143	1.3462	0.41667	0.224
Lachnospiraceae_XPB1014_group	PRJNA747262	Silva v. 138	Lefse	0.25132	0.60649	0.021053	0.022727	0	0	0.41667	0.0822
Lachnospiraceae_ND3007_group	PRJEB50040	Silva v. 138	Lefse	0.2958	0.68442	-	28	11.083	5.7424	-	1.08
Lachnospiraceae_ND3007_group	PRJEB61722	Silva v. 138	Lefse	0.34702	0.71539	-	-	-	7.8333	7.0667	−0.141
Lachnospiraceae_ND3007_group	PRJNA747262	Silva v. 138	Lefse	0.7409	0.84526	41.263	51.932	35.314	41.115	35.5	0.969
Lachnospiraceae_ND3007_group	PRJNA895415	Silva v. 138	Lefse	**0.032752**	0.35878	-	-	2.8571	0.77778	-	0.31
Lachnospiraceae_UCG_001	PRJEB50040	Silva v. 138	Lefse	0.0651	0.45349	-	1	1.2083	0.86364	-	0.0691
Lachnospiraceae_UCG_001	PRJEB61722	Silva v. 138	Lefse	**0.058343**	0.58486	-	-	-	8.5952	0.13333	−0.719
Lachnospiraceae_UCG_001	PRJNA747262	Silva v. 138	Lefse	0.15103	0.52394	9.1053	7.2955	6.7429	9.1923	1.75	0.674
Lachnospiraceae_UCG_002	PRJEB61722	Silva v. 138	Lefse	**0.016965**	0.47234	-	-	-	0	0.46667	0.0911
Lachnospiraceae_UCG_002	PRJNA747262	Silva v. 138	Lefse	0.82311	0.89087	0.021053	0.022727	0	0	0	0.00491
Lachnospiraceae_UCG_003	PRJEB50040	Silva v. 138	Lefse	0.61272	0.86598	-	0	0	0.24242	-	0.0497
Lachnospiraceae_UCG_004	PRJEB50040	Silva v. 138	Lefse	**0.019059**	0.20522	-	7.875	1.75	1.5909	-	0.617
Lachnospiraceae_UCG_004	PRJEB61722	Silva v. 138	Lefse	**0.023851**	0.47234	-	-	-	28.357	5.8	−1.09
Lachnospiraceae_UCG_004	PRJEB61722	Silva v. 138	Lefse	**0.023851**	0.47234	-	-	-	28.357	5.8	−1.09
Lachnospiraceae_UCG_004	PRJNA895415	Silva v. 138	Lefse	**0.012537**	0.21696	-	-	4.7143	1.3333	-	0.43
Lachnospiraceae_UCG_006	PRJEB50040	Silva v. 138	Lefse	**1.1569 × 10^−5^**	**0.0024526**	-	0.5	0	0	-	0.0969
Lachnospiraceae_UCG_006	PRJNA747262	Silva v. 138	Lefse	0.24373	0.60649	0.042105	0.25	0.17143	0.11538	0.75	0.132
Lachnospiraceae_UCG_010	PRJEB50040	Silva v. 138	Lefse	**0.0004688**	**0.033128**	-	19.25	4.0417	3.9848	-	0.936
Lachnospiraceae_UCG_010	PRJNA747262	Silva v. 138	Lefse	0.1599	0.52394	31.411	47.409	47.514	39	54.25	1.09
Lachnospiraceae_unclassified	PRJEB50040	Silva v. 138	Lefse	0.37074	0.76342	-	397	281	340.27	-	1.77
Lachnospiraceae_unclassified	PRJEB61722	Silva v. 138	Lefse	**0.05358**	0.58486	-	-	-	404.71	137.8	−2.13
Lachnospiraceae_unclassified	PRJNA747262	Silva v. 138	Lefse	0.77331	0.86605	524.83	499.14	536.26	492.65	452.67	1.63
Lachnospiraceae_unclassified	PRJNA895415	Silva v. 138	Lefse	0.15071	0.68777	-	-	290.14	175.8	-	1.76
Lachnospiraceae_unclassified	PRJNA684070	Silva v. 138	Lefse	**0.0035627**	**0.037853**	-	-	4001.5	4506.3	157.67	3.34

**Table 5 microorganisms-13-02061-t005:** Differential abundance of the unclassified Lachnospiraceae in Healthy vs. COVID-19 according to the re-analysis using a standard protocol. Only one study was eligible for this analysis. Bold numbers indicate statistical significance.

Genus	Study	Reference Database	Statistical Analysis	*p* Values	FDR	Healthy	COVID-19	LDA Score
Lachnospiraceae_ge	Standard Protocol	Silva v. 138	LEfSe	0.48465	0.66996	0.93333	4.0909	−0.411
Lachnospiraceae_NK3A20_group	Standard Protocol	Silva v. 138	LEfSe	0.27423	0.48637	0.93333	0.090909	0.153
Lachnospiraceae_UCG_004	Standard Protocol	Silva v. 138	LEfSe	**0.031072**	0.1211	1.1333	0	0.195
Lachnospiraceae_unclassified	Standard Protocol	Silva v. 138	LEfSe	0.26464	0.47838	8.8667	147.45	−1.85

## Data Availability

The original contributions presented in this study are included in the article. Further inquiries can be directed to the corresponding author.

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
