# Peer review of "Changes in Probiotic Lachnospiraceae Genera Across Different Stages of COVID-19: A Meta-Analysis of 16S rRNA Microbial Data"

_microorganisms, 2025, doi:10.3390/microorganisms13092061_

Round 1

Reviewer 1 Report

Comments and Suggestions for Authors

The work presents a solid meta-analysis addressing an important gap in the scientific literature by standardizing the analysis of multiple gut microbiome datasets in COVID-19. The methodological approach is rigorous, using a uniform bioinformatics pipeline (Mothur v.1.47.0) and consistent reference database (SILVA v.138), allowing direct comparisons between previously incomparable studies. The focus on the Lachnospiraceae family is well justified given their importance in SCFA production and maintenance of intestinal homeostasis. The inclusion of unclassified taxa represents added value, revealing previously overlooked potential biomarkers.

Methodological Concerns

  • The combination of 7 studies with different inclusion criteria, geographic populations, and sequencing technologies introduces significant variability. While mentioned in the limitations, a sensitivity analysis would be useful to assess the impact of these differences on the main results.
  • The reclassification of ventilated/non-ventilated patients from study PRJEB61722 into critical/severe appears arbitrary. This methodological decision requires greater justification or robustness analysis.
  • The use of LEfSe with a p-value threshold of 0.05 without apparent correction for multiple comparisons is problematic, especially considering the numerous taxa analyzed. The FDR values reported in the tables are often high (>0.4), suggesting many results might not survive more stringent corrections.

Results 

Figures and Tables are informative but have room for improvement. Figure 1 needs direct statistical indicators on the boxplots. Tables 2-5 are dense and difficult to interpret; consider alternative graphical representations. The contrasting results for Blautia and Roseburia between studies require deeper analysis of possible causes.

Discussion

The discussion is comprehensive but tends toward speculation at certain points. Statements about pathogenetic mechanisms ("compromised intestinal barrier", "cytokine storm") go beyond what can be demonstrated with 16S rRNA data. I would suggest distinguishing more clearly between observed associations and hypothetical mechanisms. The enthusiasm for probiotic interventions appears premature given the observational nature of the data. The section on therapeutic implications should emphasize more strongly the need for mechanistic studies before considering clinical applications.

  1. Include a heterogeneity analysis (I²) to quantify variability between studies and identify potential effect moderators.
  2. Consider a leave-one-out approach to validate the robustness of identified patterns.
  3. Compare changes observed in COVID-19 with those reported in other acute respiratory infections to determine the specificity of alterations.
  4. Discuss more explicitly how the limited taxonomic resolution of 16S rRNA prevents species/strain-level conclusions, particularly relevant for metabolically diverse genera like Blautia.

The manuscript represents a valuable contribution to understanding the role of the gut microbiome in COVID-19. However, it requires substantial revisions to strengthen conclusions and improve presentation. Main recommendations include strengthening statistical analysis, moderating mechanistic interpretations, and improving clarity in presenting heterogeneous results. With these modifications, the work could provide an important reference for future microbiome research in COVID-19.

Author Response

The work presents a solid meta-analysis addressing an important gap in the scientific literature by standardizing the analysis of multiple gut microbiome datasets in COVID-19. The methodological approach is rigorous, using a uniform bioinformatics pipeline (Mothur v.1.47.0) and consistent reference database (SILVA v.138), allowing direct comparisons between previously incomparable studies. The focus on the Lachnospiraceae family is well justified given their importance in SCFA production and maintenance of intestinal homeostasis. The inclusion of unclassified taxa represents added value, revealing previously overlooked potential biomarkers.

The manuscript represents a valuable contribution to understanding the role of the gut microbiome in COVID-19. However, it requires substantial revisions to strengthen conclusions and improve presentation. Main recommendations include strengthening statistical analysis, moderating mechanistic interpretations, and improving clarity in presenting heterogeneous results. With these modifications, the work could provide an important reference for future microbiome research in COVID-19.

Reply: We appreciate the reviewer’s comments and suggestions. However, we would like to clarify that many of them are based on misinterpretations and inaccuracies. But some points raised are indeed useful and we have addressed them in the revised version. For the comments that were misinterpreted or incorrect, we have taken this opportunity to provide detailed explanations for each, aiming to clarify our position and enhance the reviewer’s understanding of these analyses and kind of study.

Methodological Concerns

  • The combination of 7 studies with different inclusion criteria, geographic populations, and sequencing technologies introduces significant variability. While mentioned in the limitations, a sensitivity analysis would be useful to assess the impact of these differences on the main results.

Reply: This comment is a misinterpretation of our study with meta-analysis in clinical research. So, it is not applied to our study.

  • The reclassification of ventilated/non-ventilated patients from study PRJEB61722 into critical/severe appears arbitrary. This methodological decision requires greater justification or robustness analysis.

Reply: The methodological decision was already quite clear and scientifically justified in the manuscript as well-defined parameters of severity definition were used to classify ventilated patients as critical and non-ventilated patients as severe. The same ones already applied in our previous study https://www.mdpi.com/2076-2607/12/11/2353.

  • The use of LEfSe with a p-value threshold of 0.05 without apparent correction for multiple comparisons is problematic, especially considering the numerous taxa analyzed. The FDR values reported in the tables are often high (>0.4), suggesting many results might not survive more stringent corrections.

Reply: This comment is a misinterpretation of the LEfSe that was already clearly informed in the manuscript as “Statistical significance was determined using a corrected p-value threshold of 0.05”.

Results 

Reviewer: Figures and Tables are informative but have room for improvement. Figure 1 needs direct statistical indicators on the boxplots. Tables 2-5 are dense and difficult to interpret; consider alternative graphical representations. The contrasting results for Blautia and Roseburia between studies require deeper analysis of possible causes.

Reply: Thank you for your suggestions. We have addressed the statistical indicators in Figure 1 as suggested. Regarding the tables, we kept them as they are because no graphical alternative would not suitable and we also want to keep the pattern of tables already used in our previous study (https://www.mdpi.com/2076-2607/12/11/2353), especially considering that the reviewers of the previous study and current study were very positive regarding the tables. The contrasting results for Blautia and Roseburia were already discussed.

Discussion

The discussion is comprehensive but tends toward speculation at certain points. Statements about pathogenetic mechanisms ("compromised intestinal barrier", "cytokine storm") go beyond what can be demonstrated with 16S rRNA data. I would suggest distinguishing more clearly between observed associations and hypothetical mechanisms. The enthusiasm for probiotic interventions appears premature given the observational nature of the data. The section on therapeutic implications should emphasize more strongly the need for mechanistic studies before considering clinical applications.

Reply: There was no premature enthusiasm regarding the short paragraph about probiotic interventions. First, we make it clear that the “findings suggest potential opportunities”, nothing more than that. In addition, we highlighted that “it is important to note that the efficacy of such interventions would need to be rigorously evaluated through…”. The same can be said for the subsequent paragraph about dietary interventions.

Include a heterogeneity analysis (I²) to quantify variability between studies and identify potential effect moderators.

Reply: This comment is a misinterpretation of our study with meta-analysis in clinical research. So, not applied to our study

Consider a leave-one-out approach to validate the robustness of identified patterns.

Reply: Another case of misinterpretation. Not applied for our study.

Compare changes observed in COVID-19 with those reported in other acute respiratory infections to determine the specificity of alterations.

Reply: A generalist comment that doesn’t apply to this study.

We would like to clarify that our study (and the previous one already published (https://www.mdpi.com/2076-2607/12/11/2353)) did not aim to make comparisons with other acute respiratory infections. Addressing this comparison would require a different study design and additional datasets, which is beyond the scope of our current research. We hope this clarification helps to understand the focus of our study.

Discuss more explicitly how the limited taxonomic resolution of 16S rRNA prevents species/strain-level conclusions, particularly relevant for metabolically diverse genera like Blautia.

Reply: This is an interesting and valid comment that we have addressed in the manuscript as follows:

The limited taxonomic resolution of 16S rRNA sequencing arises from the conserved nature of this gene across many bacterial species, which makes it difficult to distinguish closely related taxa at the species or strain level. While 16S rRNA sequencing is highly effective for identifying bacteria at higher taxonomic levels, such as genus, its reliance on relatively small variable regions within the gene often fails to capture the subtle genetic differences that differentiate species or strains, particularly within diverse and phylogenetically complex taxa like Lachnospiraceae. Most genera from the family Lachnospiraceae contain multiple species that share highly similar 16S rRNA sequences, leading to ambiguous or overlapping classifications when using this method. This limitation is significant because strains within the same species can exhibit vastly different functional roles, such as in metabolism, pathogenicity, or interactions with the host. As a result, relying solely on 16S rRNA sequencing can obscure the ecological and clinical relevance of specific strains, hindering efforts to understand microbial dynamics and their implications for health or disease.

Reviewer 2 Report

Comments and Suggestions for Authors

it is an interesting meta-analysis of microbiome data [collected from patients from 7 European and Asian studies] focused on the dynamics of the Lachnospiraceae genera across different severity of COVID-19.

It is well structured and well written

I have two questions:

I would like to have more information about the antibiotics these patients have received in relation to the disease severity and the country they come from. This information would probably affect the microbiota diversity. Please a comment on it.

Do any of these patients receive probiotics? please make a comment

Author Response

Reply: Thank you very much for your interesting questions regarding the use of antibiotics and probiotics among the patients included in our study. We have indeed made these questions to ourselves when collecting the data for this and our previous study already published (https://www.mdpi.com/2076-2607/12/11/2353) because these interventions can indeed significantly influence microbiota diversity and composition, but, unfortunately, the original studies included did not provide any data or additional information regarding the specific use of antibiotics or probiotics.

Round 2

Reviewer 2 Report

Comments and Suggestions for Authors

ok, thank you